# I Don't Know: Explicit Modeling of Uncertainty with an `[IDK]` Token

**Roi Cohen**
HPI / University of Potsdam
Roi.Cohen@hpi.de

**Konstantin Dobler**
HPI / University of Potsdam
Konstantin.Dobler@hpi.de

**Eden Biran**
Tel Aviv University
edenbiran@mail.tau.ac.il

**Gerard de Melo**
HPI / University of Potsdam
Gerard.DeMelo@hpi.de

## Abstract

Large Language Models are known to capture real-world knowledge, allowing them to excel in many downstream tasks. Despite recent advances, these models are still prone to what are commonly known as hallucinations, causing them to emit unwanted and factually incorrect text. In this work, we propose a novel calibration method that can be used to combat hallucinations. We add a special `[IDK]` ("I don't know") token to the model's vocabulary and introduce an objective function that shifts probability mass to the `[IDK]` token for incorrect predictions. This approach allows the model to express uncertainty in its output explicitly. We evaluate our proposed method across multiple model architectures and factual downstream tasks. We find that models trained with our method are able to express uncertainty in places where they would previously make mistakes while suffering only a small loss of encoded knowledge. We further perform extensive ablation studies of multiple variations of our approach and provide a detailed analysis of the precision-recall tradeoff of our method.[1]

## 1 Introduction

Large Language Models (LLMs) are pretrained on massive amounts of text to understand and generate language. This training text includes a large portion of written human knowledge such as books, newspapers, Wikipedia, and scientific articles. During this process, LLMs also retain a remarkable amount of the information seen during pre-training, allowing them to encode real-world knowledge in their parameters and act as knowledge bases [Petroni et al., 2019, Roberts et al., 2020, Cohen et al., 2023a, Pan et al., 2023]. Owing to this phenomenon, LLMs can be used in multiple settings requiring this real-world knowledge, such as closed-book question answering [Brown et al., 2020, Roberts et al., 2020] and information retrieval [Tay et al., 2022].

Despite the popularity of LLMs, they are prone to what is commonly referred to as hallucinations, which severely hinder their performance and reliability [Ji et al., 2023, Manduchi et al., 2024]. Examples of hallucinations include factually incorrect [Maynez et al., 2020, Devaraj et al., 2022, Tam et al., 2023], inconsistent [Elazar et al., 2021, Mündler et al., 2023], self-contradicting [Cohen et al., 2024] or non-attributable text [Bohnet et al., 2022, Rashkin et al., 2023, Yue et al., 2023].

A prominent method employed to combat such hallucinations is model calibration [Guo et al., 2017a, Brundage et al., 2020], which aims to calibrate the confidence of model predictions such that they

---

[1]We release our code and IDK-tuned model checkpoints at https://github.com/roi-hpi/IDK-token-tuning.

38th Conference on Neural Information Processing Systems (NeurIPS 2024).

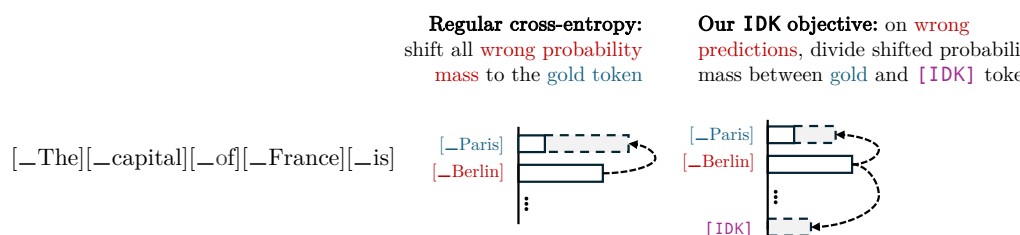

Figure 1: Illustration of our proposed IDK objective. During continual pretraining, we shift some probability mass of wrong predictions towards a special [IDK] token. The amount of shifted probability mass depends on the uncertainty in the model's prediction. We detail our method in Section 2.

are better aligned with their quality. This calibration allows LLMs to explicitly express uncertainty, allowing them to caveat their responses or even refrain from answering. Although many of the proposed methods do lead to an improvement in model calibration [Geng et al., 2024], they have still been found to be lacking [Chen et al., 2023].

In this work, we propose a novel objective function that allows LLMs to explicitly express uncertainty. We add a new special [IDK] ("I Don't Know") token to the vocabulary of the language model. During a continued pretraining phase, we modify the conventional cross-entropy objective to express uncertainty in a next-token prediction as probability mass on the [IDK] token. Specifically, each time the model fails to predict the gold label, some of the probability mass of the target is shifted to the [IDK] token based on an *Uncertainty Factor* we calculate based on the predicted logits. We refer to our method as IDK-tuning.

Our proposed IDK objective differs from previous work as we intervene during a continued pretraining phase with the language modeling task. Crucially, we do not rely on any labeled data. Moreover, this allows the model to be later finetuned on specific tasks while the model has already learned to express uncertainty.

We conduct IDK-tuning using various model architectures and sizes, and then evaluate them on diverse factual downstream tasks. Our results show a large increase in factual precision of IDK-tuned models while causing only a small decrease in recall of factual knowledge that was contained in the base model. We conduct extensive ablation studies for the individual components of our IDK objective and analyze its effect on optimization dynamics. We finally show that IDK-tuning does not harm the general language modeling ability of models, such as long text generation.

In summary, our contributions include:

- We propose a novel IDK objective applied during pretraining which models uncertainty in a model's prediction as probability mass put on a special [IDK] token.
- We evaluate our objective using a large range of base models with different architectures and model sizes, and confirm the efficacy of IDK-tuning on a range of factual answering downstream tasks.
- We extensively analyze individual components of our objective and its effect on general language modeling ability.

## 2  IDK-tuning

Our goal is to train a model to be aware of its unawareness and to effectively express it. For this, we introduce a new special token to its vocabulary: [IDK]. The model is intended to express uncertainty by putting probability mass on the [IDK] token in its predictions. In practice, we adapt the model's pretraining objective, aiming to teach it to use the [IDK] token effectively. Our objective does not require annotations of uncertainty or specifically crafted datasets (e.g., Q&A). Instead, we leverage the uncertainty captured by the pretraining objective on its pretraining data and use it to encourage probability mass on the [IDK] token in cases of uncertainty. We hypothesize that this generalizes to uncertainty expressed on downstream tasks like Q&A, which we experimentally verify later on.

We next describe in detail the technicalities of the [IDK] token and our training method.

## 2.1 The `[IDK]` token

The purpose of the new token is to represent lack of knowledge. Ideally, whenever the model would have been making a mistake, we want it to instead predict this token. That is, rather than generating a wrong token, we would like to model to generate the `[IDK]` token, as a means of conveying its uncertainty. We can consider this as a model expressing its lack of knowledge and may then choose to ignore its outputs. The more the model opts for this token rather than predicting the wrong answer, the more we improve the model's precision.

For instance, let us consider the setup of Factual Sentence Completion. In this setup, the model receives an incomplete sentence as an input and is expected to complete it factually. For example, a valid input would be "`Paris is the capital of`", and a factually correct output by the model would be "`France`". In this setup, if the model was going to predict "`Germany`", using the `[IDK]` token instead increases factual precision by refusing to answer a question where the answer would have been wrong. Naturally, almost universally predicting `[IDK]` indiscriminately may yield high precision but is not helpful. Therefore, taking into account the recall of factually correct answers is crucial in evaluating our method. We analyze both the precision and recall of our method in Section 4.

We add this new `[IDK]` token to the model's vocabulary and initialize its embedding randomly. The embedding is optimized alongside the rest of the model's parameters during training. We next describe our proposed IDK objective.

## 2.2 The `IDK` Training Objective

We modify the conventional cross-entropy objective between the softmax distribution over the model's prediction and the correct answer, such that each time the model fails to predict the correct token, it is encouraged to instead put some probability mass on the `[IDK]`. This encouragement is modulated by an *Uncertainty Factor* denoted as $\lambda \in [0, 1]$ that is larger the more uncertain the model is and exactly 0 when the model predicts the correct token.

We now define our modified cross-entropy objective. We use `[gold]` to denote the gold token (correct target) for each prediction. We denote the probability mass assigned to an arbitrary token `[tok]` in the prediction of a model as $\text{prob}(y_t = \texttt{[tok]}|y_{<t}, x)$ We further use $\mathbf{1}_{\texttt{[IDK]}}$ to denote a one-hot target vector with one at the index of the `[IDK]` token. Per convention, $\mathbf{y}$ denotes the one-hot target vector for the `[gold]` token. The modified objective is defined as follows:

$$\mathcal{L}_{\text{IDK}} = \mathcal{L}_{\text{CE}}(\hat{\mathbf{y}}, (1 - \lambda)\,\mathbf{y} + \lambda\,\mathbf{1}_{\texttt{[IDK]}}) \tag{1}$$

If the model is uncertain in its prediction, the target is shifted away from predicting the `[gold]` token and towards the `[IDK]` token. This is modulated by $\lambda$. Note that in case the model makes the correct prediction, $\lambda = 0$ and $\mathcal{L}_{\text{IDK}}$ therefore reduces to the regular cross-entropy loss. When the model is correct, $\mathcal{L}_{\text{IDK}}$ simply provides the signal for the correct prediction. When the model is incorrect, $\mathcal{L}_{\text{IDK}}$ provides both the signal for the correct prediction and a signal to express uncertainty. We now detail the construction of the *Uncertainty Factor* $\lambda$.

**The *Uncertainty Factor*.** $\lambda$ is constructed as a scalar weight with $\lambda \in [0, 1]$. Intuitively, we want $\lambda$ to be close to 1 when the model is very uncertain and 0 when the model makes the correct prediction. Based on this, we define $\lambda$ as one minus the probability mass on the gold token divided by the maximum probability mass put on any token:

$$\lambda = \Pi \times \left(1 - \frac{\text{prob}(y_t = \texttt{[gold]}|y_{<t}, x)}{\max_i(\text{prob}(y_t = i|y_{<t}, x))}\right), \tag{2}$$

where $\Pi \in [0, 1]$ is a hyperparameter to control the influence of our objective. When the gold token probability is close to the maximum probability, $\lambda$ is close to 0. If the model makes a correct prediction (the gold token is assigned the maximum probability), $\lambda$ is 0, thereby reducing Equation 1 to the regular cross-entropy loss. When the gold token probability is much lower than the maximum probability, $\lambda$ is close to 1, which translates to shifting almost all the probability mass of the target in Equation 1 to the `[IDK]` token. $\Pi$ specifies the upper bound of target probability mass that can be shifted to the `[IDK]` token. For example, $\Pi = \frac{1}{2}$ means that at most half of the probability mass

in the target can be shifted to [IDK] while the rest remains with the gold token. In practice, we do not tune this and set $\Pi = \frac{1}{2}$. This prevents the [IDK] token from ever becoming a better prediction than the gold token while still providing enough signal to predict [IDK] for uncertain predictions. We perform an ablation of the influence of $\Pi$ in Section 4.2.

**Uncertainty Regularization.** An important consideration in designing the $\mathcal{L}_{\text{IDK}}$ objective is to prevent a collapse where the model is miscalibrated with too many false positive [IDK]s, putting too much probability mass on [IDK], although it could have made the correct prediction. Therefore, we add the following anti-false positive regularization to our objective:

$$\mathcal{L}_{\text{FP-reg}} = -\log(1 - \texttt{prob}(y_t = \texttt{[IDK]}|y_{<t}, x)), \tag{3}$$

which is exactly the binary cross-entropy objective with 0 as the target and the probability mass assigned to the [IDK] as the input. We only add this regularization objective when the model's prediction is correct. This aims to minimize the [IDK] token's probability mass the model learns to predict in cases it knows the answer – thus teaching it to minimize the use of this token in cases it is more certain, and is designed to reduce a decrease of its recall. We perform an ablation of $\mathcal{L}_{\text{FP-reg}}$ in Section 4.2.

**The final loss.** Combining all objectives, our final IDK objective is therefore:

$$\mathcal{L} = \begin{cases} \mathcal{L}_{\text{CE}} + \mathcal{L}_{\text{FP-reg}} & \text{if } \lambda = 0 \\ \mathcal{L}_{\text{IDK}} & \text{otherwise.} \end{cases} \tag{4}$$

## 3 Experiments

We use our proposed IDK objective to tune various pretrained models to use the new [IDK] token. We dub this process IDK-tuning. We then report the results of the IDK-tuned models on commonly used factual benchmarks, showing that our method improves factuality while paying only a small price in terms of knowledge recall. We also show that model size plays a significant role in the success of our method to create an effective uncertainty-aware model.

We employ *continual training* of pretrained models rather than training from scratch for two reasons: (i) the computational cost of training models that perform competitively on current benchmarks from scratch would be prohibitive, and (ii) starting from a model that is already a strong language modeler helps during the optimization process by providing a rough initial calibration that we utilize to derive the *Uncertainty Factor*.

### 3.1 IDK-tuning Setup

We use bert-base-cased [Devlin et al., 2019], mistralai/Mistral-7B-v0.1 [Jiang et al., 2023], and EleutherAI/pythia-70m − 2.8B [Biderman et al., 2023] for our base models for IDK-tuning. For IDK-tuning Mistral-7B-v0.1, we train on data randomly sampled from The Pile [Gao et al., 2020][2] with a context length of 4,096. We use example packing to fill the entire context length. We use a maximum learning rate of $4 \times 10^{-5}$ with a linear warmup for 10% of the training steps and a cosine decay down to $2 \times 10^{-6}$. We use a batch size of 256, weight decay of 0.05, gradient clipping of 1.0 and AdamW betas (0.9, 0.95). We train for 1,024 optimizer steps resulting in a total of 1B training tokens. For the pythia-70m − 2.8B models, we use the same hyperparameters but reduce the context length to 2,048 to match the model's positional embeddings. We use bfloat16 and float16 mixed-precision training to match Mistral-7B-v0.1 and pythia-410m − 2.8B pretraining, respectively. For pythia-70m, pythia-160m and bert-base-cased, we observed NaN errors in the predicted logits irrespective of our loss modifications. Since the models are small enough, we switch to pure float32 for these models without using mixed-precision. In addition, for bert-base-cased we apply MLM [Devlin et al., 2019], while for each input, we randomly mask one of the tokens.

---

[2]We use monology/pile-uncopyrighted on the Huggingface Hub for a version of The Pile without the Books corpus, which contains copyrighted works.

## 3.2 Evaluation Setup

**Evaluation Data.** We consider the following datasets: LAMA [Petroni et al., 2019], TriviaQA [Joshi et al., 2017], and PopQA [Mallen et al., 2022]. These cover a wide range of queries, for example trivia questions (TriviaQA), and subject-relation-object facts phrased as queries (LAMA, PopQA). We consider the closed-book open-ended setting, where we do not provide any context or answer choices to the model. Importantly, in the case of TriviaQA and PopQA, where the input is formed as a question, we reduce it into a sentence completion task, using GPT4. Specifically, we prompt it to phrase the question as a sentence, while also providing it with some in-context examples that we manually created. See Appendix C for more details and the full prompt. To evaluate multiple-choice question answering, we use EleutherAI's `lm-evaluation-harness` [Gao et al., 2023]. Specifically, we use ARC [Clark et al., 2018], HellaSwag [Zellers et al., 2019], MMLU [Hendrycks et al., 2020], TruthfulQA [Lin et al., 2022a], WinoGrande [Sakaguchi et al., 2021], and GSM8k [Cobbe et al., 2021].

**Baselines.** For each of the evaluation datasets, we compare the `IDK`-tuned model with its original base model without any further training. Furthermore, we consider three different baselines:

1. **Confidence Threshold** baseline: We use the predicted probability mass in the language modeling head of the LM as a measure of confidence in the prediction [Yoshikawa and Okazaki, 2023]. We consider the first token generated by the LM. In case the corresponding probability mass of this token is greater than a fixed threshold, we consider the generation as valid. Otherwise, we consider this as an uncertainty expression (analogous to an `[IDK]` token generation in our model). To create a strong baseline, we search for the best threshold via hyperparameter tuning on the development set.

2. **P(True)** baseline [Kadavath et al., 2022]: Given an input sentence to complete, which we refer to as $I$, we use the original model to generate the completion, which we refer to as $A$. We then concatenate $I$ and $A$ and ask the model: *"Please answer either with 'true' or 'false' only. Is it true that: $I A$"*. If the model answer is not 'true', we consider this specific example as unknown for the model – namely the same case as if the IDK-tuned model would generate the `[IDK]`.

3. **Semantic Entropy** baseline [Kuhn et al., 2023, Aichberger et al., 2024]: We sample $K$ text generations from the model, encode them using a state-of-the-art semantic encoder and cluster their encodings. If the largest cluster size is larger than $\frac{K}{2}$, then we take a random generation out of this cluster as the model's answer. Otherwise, we consider this example as unknown.

**Evaluation.** We evaluate how well our models use the new `[IDK]` token by measuring their factuality and knowledge memory, using the following metrics: (i) **Precision**: the portion of factually correct completions, out of all the claims that have been completed with any token that is different from the `[IDK]` token, i.e., the claims that the model was certain enough about, and tried to factually complete. (ii) **Recall**: the portion of factually correct completions, out of all the claims in the dataset. Namely, the portion of knowledge memory the model has, out of the entire test set we evaluate on. (iii) **F1**: the harmonic mean of precision and recall. In the case of base models without additional calibration methods, the precision, recall, and F1-scores all correspond to their accuracy on the task.

In Section 4.2, we use two further metrics to analyze the patterns when IDK-tuned models predict `[IDK]`. For this, we use the notion of correctly predicting `[IDK]`: We consider an `[IDK]` prediction to be correct if the base model does not predict the correct answer for an instance. We define (i) **IDK recall**: the fraction of instances the model predicted `[IDK]` correctly out of all instances where the base model did in fact not predict the correct answer, and (ii) **IDK error rate**: the fraction of instances where the model predicted `[IDK]` incorrectly out of all instances where the base model did indeed predict the correct answer.

## 4 Results

We next report results showing that our proposed `IDK`-tuning method can effectively improve factuality while causing only a small loss of existing knowledge.

| | LAMA Google-RE | | | LAMA T-Rex | | | LAMA SQuAD | | | TriviaQA | | | PopQA | | |
|---|---|---|---|---|---|---|---|---|---|---|---|---|---|---|---|
| | **P** | **R** | **F1** | **P** | **R** | **F1** | **P** | **R** | **F1** | **P** | **R** | **F1** | **P** | **R** | **F1** |
| `Mistral-7B-v0.1` | 48.1 | 48.1 | 48.1 | 71.2 | **71.2** | 71.2 | 45.8 | 45.8 | 45.8 | 52.0 | 52.0 | 52.0 | 35.5 | **35.5** | **35.5** |
| `Mistral-7B-v0.1` + The Pile | 48.8 | **48.8** | 48.8 | 69.9 | 69.9 | 69.9 | 48.0 | **48.0** | 48.0 | 52.2 | **52.2** | 52.2 | 35.2 | 35.2 | 35.2 |
| `Mistral-7B-v0.1` + Confidence Threshold | 60.0 | 40.0 | 48.0 | 80.4 | 63.5 | 71.0 | 64.4 | 33.5 | 44.1 | 70.4 | 41.1 | 51.9 | 64.6 | 20.6 | 31.2 |
| `Mistral-7B-v0.1` + P(True) | 54.4 | 44.5 | 48.9 | 73.8 | 65.1 | 69.2 | 54.9 | 41.0 | 46.9 | 58.8 | 47.5 | 52.5 | 40.3 | 29.0 | 33.7 |
| `Mistral-7B-v0.1` + Semantic Entropy | 70.1 | 38.9 | 50.0 | 88.0 | 65.4 | 75.0 | 70.2 | 44.5 | 54.4 | 68.5 | 52.5 | 59.4 | 68.7 | 20.4 | 31.5 |
| **`Mistral-7B-v0.1` + IDK-tuning** on The Pile | **71.1** | 40.6 | **51.7** | **88.5** | 65.5 | **75.3** | **72.0** | 44.3 | **54.9** | **72.5** | 52.0 | **60.6** | **78.1** | 20.5 | 32.5 |

Table 1: Precision (P), Recall (R), and F1-scores for `Mistral-7B-v0.1`. Our IDK-tuning achieves the best precision with minor decreases in recall, outperforming previous work. `Mistral-7B-v0.1` + Confidence Threshold refers to the baseline based on the probability mass of the predicted answer [Yoshikawa and Okazaki, 2023]. `Mistral-7B-v0.1` + The Pile refers to the ablation discussed in Section 4.2.

| | **P** | **R** | **F1** |
|---|---|---|---|
| `Mistral-7B-v0.1` | 28.2 | **28.2** | 28.2 |
| `Mistral-7B-v0.1` + The Pile | 28.3 | 28.3 | 28.3 |
| `Mistral-7B-v0.1` + Confidence Threshold | 45.0 | 18.5 | 26.2 |
| **`Mistral-7B-v0.1` + IDK-tuning** on The Pile | **48.8** | 20.8 | **29.2** |

Table 2: Precision (P), Recall (R), and F1-scores of our model on the `lm-eval-harness`, compared to baselines.

## 4.1 Main Results

**`Mistral-7B-v0.1` results.** Table 1 shows the results of our largest model `Mistral-7B-v0.1` on factual closed-book sentence completion datasets. Our results show that the IDK-tuned `Mistral-7B-v0.1` has a much higher precision – namely the model generates significantly fewer factually incorrect completions and instead puts probability mass on the `[IDK]` token. However, the model does show decreased knowledge recall on some tasks. Overall, we observe an increase in the average F1-score. Table 2 shows the averaged results on the `lm-eval-harness` datasets. The trend here is similar, although the increase in precision compared to baselines is slightly lower. This suggests that the model tends to be more certain when it comes to multiple-choice questions.

**Scaling behavior of IDK-tuning.** We further investigate the effect of model size on the success of IDK-tuning. We conduct IDK-tuning for each of the `pythia-70m − 2.8B` models as detailed in Section 3.1. In Figure 2, we plot the average precision, recall, and F1-score for each of `pythia-70m − 2.8B` as well as `Mistral-7B-v0.1`, over all the closed-book sentence completion datasets. We observe a clear trend of recall and F1-score increasing log-linearly with the model size. The precision of IDK-tuned models increases only slightly as the model size increases. For the two smallest models we investigate (`pythia-70m` and `pythia-160m`), our method is arguably not effective, as the IDK-tuned model's recall collapses (we further analyze this in Section 4.3).

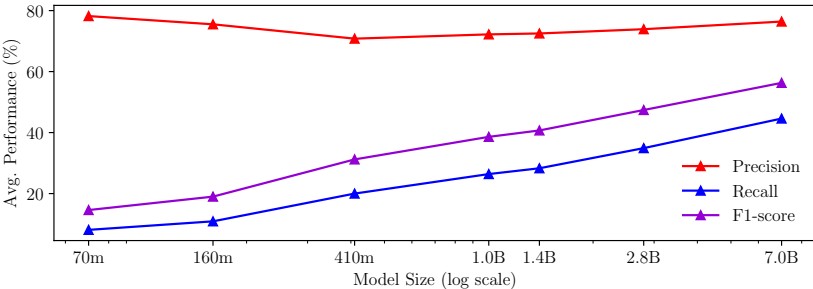

Figure 2: Average performance on closed-book factual sentence completion benchmarks of IDK-tuned models in terms of their parameter count. 70m to 2.8B are `pythia-70m − 2.8B`, while 7.0B is `Mistral-7B-v0.1`.

|  | LAMA Google-RE | | | LAMA T-Rex | | | LAMA SQuAD | | |
|---|---|---|---|---|---|---|---|---|---|
|  | **P** | **R** | **F1** | **P** | **R** | **F1** | **P** | **R** | **F1** |
| bert-base-cased | 23.0 | **23.0** | 23.0 | 59.8 | **59.8** | 59.8 | 9.5 | **9.5** | 9.5 |
| bert-base-cased + Confidence Treshold | 58.8 | 15.8 | 24.9 | 71.5 | 35.9 | 47.8 | 69.5 | 5.0 | 9.3 |
| **bert-base-cased + IDK-tuning** | **78.1** | 15.9 | **26.4** | **72.5** | 53.0 | **61.2** | **80.2** | 6.4 | **11.9** |

Table 3: Precision (P), Recall (R), and F1 scores for of our IDK-tuned `bert-base-cased` on the evaluation benchmarks, compared to baselines.

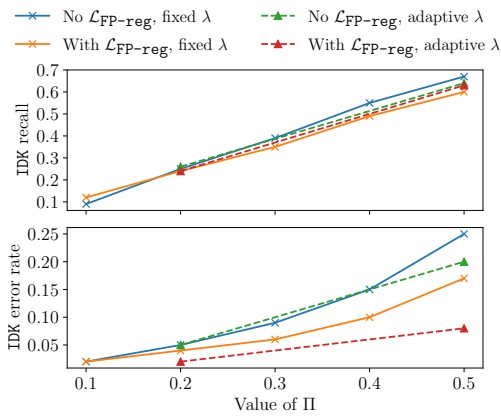

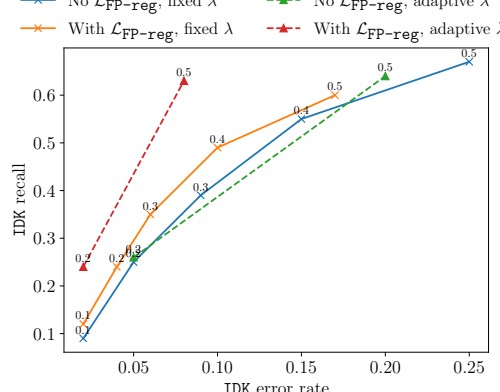

Figure 3: Ablation study of different values for the $\Pi$ factor that controls the upper bound of probability mass put on [IDK] in the target.

Figure 4: Tradeoff between IDK recall and IDK error rate for different parameter combinations. We annotate each data point with its corresponding $\Pi$ value.

**bert-base-cased results.** Table 3 reports the results of out IDK-tuned `bert-base-cased` model. We see a similar trend as in our evaluation of `Mistral-7B-v0.1`. Factuality is improved, while recall is reduced by only a small amount.

## 4.2 Ablations

We perform an ablation study of our method to further investigate the effectiveness of each of its components. For our study, we calculate the IDK recall and IDK error rate on the closed-book factual sentence completion datasets. We study the effect of $\Pi$, $\lambda$ and the $\mathcal{L}_{\text{FP-reg}}$ term. For this, we perform IDK-tuning using `Mistral-7B-v0.1` with the same hyperparameters as our main runs with different combinations of the studied components[3]. We plot the IDK recall for different values of $\Pi$ in Figure 3. In Figure 4, we plot the IDK recall vs. IDK error rate tradeoff. IDK recall and IDK error rate are defined in Section 3.2. We study different aspects of these results below:

**1. Analysis of the adaptive nature of the *Uncertainty Factor* $\lambda$.** The *Uncertainty Factor* $\lambda$ defined in Equation 2 is *adaptive*, meaning the amount of probability mass shifted to [IDK] depends on the predicted probability distribution. Another possible choice is to use a *fixed* $\lambda \in [0, 1]$. We analyze this in Figure 3 and Figure 4[4]. We can see that using the adaptive $\lambda$ formulation results in a lower IDK error rate without a major decrease in IDK recall.

**2. Effect of the $\mathcal{L}_{\text{FP-reg}}$ regularization.** We also study the effect of the $\mathcal{L}_{\text{FP-reg}}$ term (see Section 2.2). Again, we see that using $\mathcal{L}_{\text{FP-reg}}$ results in a reduced IDK error rate without decreasing IDK recall significantly.

---

[3]Due to computational constraints, we run this for a reduced set of $\Pi$ for the cases with adaptive $\lambda$.
[4]For a fixed $\lambda$, we set $\lambda = \Pi$.

**3. Effect of the upper bound hyperparameter $\Pi$.**    We also study the effect of $\Pi$, which is the upper bound of the *Uncertainty Factor* (see Equation 2). Our ablation study demonstrates that increasing $\Pi$ results in an increase in correct predictions of [IDK] (higher IDK recall), at the cost of a small increase of erroneous [IDK] predictions (IDK error rate). The IDK error rate increases less when using both our proposed adaptive $\lambda$ and $\mathcal{L}_{\texttt{FP-reg}}$.

**Effect of knowledge contained in The Pile.**    Since we conduct further pretraining on The Pile, improved performance of our method could be partly explained by additional knowledge that the model learns during IDK-tuning. However, we show that this is not the case. In the case of the `pythia-70m` − 2.8B models, our data used for IDK-tuning exactly matches their pretraining data. For `Mistral-7B-v0.1`, this is not known although The Pile was likely also included. We note that the language modeling performance on The Pile of our models during IDK-tuning actually very slightly decreases rather than improving, suggesting the absence of any newly learned knowledge. However, to completely rule out any such effects, we trained `Mistral-7B-v0.1` on the exact sample of The Pile used for IDK-tuning but with the regular cross-entropy objective. We report the performance of this model in Table 1. Indeed, `Mistral-7B-v0.1` with further training on The Pile performs similarly to the base `Mistral-7B-v0.1` on average.

## 4.3    Analysis of Optimization Stability

**Collapse to [IDK].**    Highly optimizing every component of the standard language modeling task with Transformers has made it easy to forget that optimization processes of deep neural networks can be brittle and divergent. Naively replacing the regular cross-entropy objective with our IDK-loss $\mathcal{L}_{\texttt{IDK}}$ leads to a collapse of training where the model simply always learns to put most probability mass on the [IDK]. We already account for this by (i) introducing the $\Pi$ inhibitor to allow us to set an upper bound on the maximum probability mass that is assigned to the [IDK] in the target vector and (ii) introducing the additional $\mathcal{L}_{\texttt{FP-reg}}$ regularization to provide an additional signal that punishes probability mass being assigned to [IDK] when the model's prediction is already correct.

In practice, we see that the regular cross-entropy loss shows a small uptick at the very beginning of IDK-tuning. In almost all runs, this recovers quickly back to baseline levels, where it remains. We find that with $\Pi = 0.5$ and the $\mathcal{L}_{\texttt{FP-reg}}$ regularization, most training runs are stable without further model-specific tuning.

**Collapse for small models `pythia-70m` and `pythia-160m`.**    However, for `pythia-160m` and `pythia-70m`, which are the only runs in our experiments that diverge even with our added regularization losses, the regular cross-entropy keeps on rising with a large spike. Concretely, the predicted distributions not only show an increased cross-entropy with the targets but also a sharply increasing entropy: we observe that the predicted distributions collapse towards a uniform distribution. At the worst point, 0% of the predictions of `pythia-160m` are correct. However, both models somewhat recover towards the end of training but stay well below baseline levels in terms of language modeling performance. We note that this is a different collapse pattern than the collapse towards almost always predicting [IDK] observed without our regularization terms.

We further analyzed this and observe that for both `pythia-160m` and `pythia-70m`, the initial probability mass assigned to the [IDK] token is so small that it gets rounded to zero even when using `float32` precision. This causes the $\mathcal{L}_{\texttt{IDK}}$ loss to be very large, resulting in large gradient norms. Already for `pythia-410m`, the initial probability mass on [IDK] is substantial enough to prevent this (albeit still a very small value smaller than $5 \times 10^{-9}$). Both `pythia-160m` and `pythia-70m` also show a larger initial entropy in their predicted distributions (i.e., "flatter" predicted distributions). We conjecture that an adapted initialization of the [IDK] token and/or a small bias towards [IDK] at the beginning of training could prevent this divergence. As we only encounter this issue for the small `pythia-160m` and `pythia-70m` models, we leave further investigation of this for future work.

## 4.4    Text Generation

To assess whether our IDK-tuning might harm other different downstream language skills, which are not necessarily only factual, we evaluate the IDK-tuned `Mistral-7B-v0.1` on the task of text summarization, and compare its results to those of the original model. For this, due to the high likelihood of the [IDK] token being generated during a longer text generation process, we use greedy decoding and ignore the [IDK] token. For this experiment, we use four different common

|  | Legal Plain English | TLDR | SPEC5G |
|---|---|---|---|
| `Mistral-7B-v0.1` | **17.5** | **14.1** | 37.2 |
| `Mistral-7B-v0.1` + The Pile | 17.4 | **14.1** | **37.3** |
| **`Mistral-7B-v0.1` + IDK-tuning** on The Pile | 17.2 | 14.0 | 36.9 |

Table 4: RougeL scores on different summarization tasks to measure the impact of `IDK`-tuning on other language model abilities. `Mistral-7B-v0.1` + The Pile refers to the ablation discussed in Section 4.2.

|  | No effect | Noise | White Noise | Abstaining |
|---|---|---|---|---|
| `Mistral-7B-v0.1` | 68.5% | 9% | 6.5% | 16% |
| `pythia-2.8B` | 59.5% | 13.5% | 12.5% | 14.5% |
| `pythia-70m` | 52% | 18.5% | 22% | 7.5% |

Table 5: Error type distribution on 200 failures of our IDK-tuned models.

summarization benchmarks: Legal Plain English [Manor and Li, 2019], TLDR [Völske et al., 2017], and SPEC5G [Karim et al., 2023]. We measure performance using RougeL [Lin, 2004], as it is widely used in related work, and report the results in Table 4. The `IDK`-tuned `Mistral-7B-v0.1` performs only slightly worse than the original base model. This is an encouraging result, as it means that `IDK`-tuning does not necessarily harm other language skills of pretrained language models.

### 4.5 Error Analysis

To gauge the effect of `IDK`-tuning on model responses to factual prompts and questions, we conduct an in-depth manual analysis on a random sample of 200 (40 from each dataset) of the model's incorrect generations (generations that do not contain the correct answer). We conduct this analysis for three models across model sizes: `pythia-70m`, `pythia-2.8B`, and `Mistral-7B-v0.1`. We then categorize each of these incorrect generations to one of the following categories:

1. *No Effect*: Both the original model and the `IDK`-tuned model generate the same (incorrect) answer.
2. *Noise*: The original model generates the correct answer, while the `IDK`-tuned model does not.
3. *White Noise*: Both the original and `IDK`-tuned models do not generate the correct answer, however the `IDK`-tuned model generates a different one.
4. *Abstain*: The `IDK`-tuned model abstains from answering by generating text such as "un-known" or "mystery".

The results are shown in Table 5. Our analysis suggest that first, the bigger the model, the fewer changes our training approach causes in the model's generations, and second, the bigger the model, the greater its ability to abstain from answering via words (which generally can be interpreted as equal to generating an `[IDK]` token, although harder to evaluate automatically).

## 5 Related Work

**Model Calibration.** Our goal is closely related to the key challenge of model calibration [Guo et al., 2017b]: to provide a measure of the probability that a prediction is incorrect alongside the actual prediction. The problem of factual error detection can be viewed as a variation of calibration, where instead of a continuous probability, we provide a binary prediction for whether the model is correct or not. This is also related to the setting of selective prediction, where models can abstain from answering a query [Varshney et al., 2022, Kamath et al., 2020]. Common approaches to calibration are to perform various transformations on a model's output logits [Desai and Durrett, 2020, Jiang et al., 2021], and measuring uncertainty [e.g., see Kuhn et al., 2023]. More recent works have studied the use of LMs for providing calibration, by training them on statements known to be factually

correct or incorrect. This "supervised" approach has been explored via fine-tuning [Kadavath et al., 2022, Lin et al., 2022b], in-context learning [Cohen et al., 2023a, Alivanistos et al., 2022], zero-shot instruction-oriented [Cohen et al., 2023b] and consistency sampling [Yoran et al., 2023] techniques. Further recent studies [Azaria and Mitchell, 2023] use the internal state of the model for classifying whether it is certain or not, use a new token for unanswerable inputs [Lu et al., 2022], or construct a specific dataset for effectively tuning the model for answering refusal [Zhang et al., 2024]. Our work builds upon this, aiming to teach the model to assess and express its own uncertainty via the new [IDK] token we introduced.

**Attribution.** Another related line of work focuses on checking whether LM-generated texts are faithful to a given source text [Bohnet et al., 2022, Honovich et al., 2022]. This problem has been addressed via several approaches, including question generation [Wang et al., 2020, Honovich et al., 2021, Scialom et al., 2021], NLI [Thorne et al., 2018, Welleck et al., 2019, Maynez et al., 2020, Dziri et al., 2022, Gao et al., 2022, Kamoi et al., 2023], data augmentation [Atanasova et al., 2022, Wright et al., 2022, Gekhman et al., 2023], and planning schemes that allow the model to self-edit its own generation [Schick et al., 2022]. Unlike these works, we are not assuming any reference text or external knowledge bases. Instead, we aim to teach the model to decide on its own whether it is likely to be able to factually complete a sentence correctly.

# 6 Conclusion

We propose a novel method for improving LMs' factuality by adding a special [IDK] token to an LM's vocabulary. Alongside the new [IDK] token, we introduce a novel pretraining objective called IDK-tuning to model uncertainty in the model's prediction as the probability mass assigned to the [IDK]. Crucially, IDK-tuning requires no labeled data and is instead a drop-in replacement of the conventional cross-entropy loss used for self-supervised language modeling on web-crawled texts. This allows us to explore uncertainty-aware training at a large scale. In our experiments, we conduct continued pretraining of a diverse range of pretrained models using the IDK objective.

Evaluation on factual sentence completion and multiple-choice benchmarks shows that IDK-tuned models can complete these tasks with much higher precision by refusing to answer (assigning high probability mass to the [IDK] token) in cases when the base model would have given a wrong answer. This comes at only small decreases in recall. We investigate the scaling behavior of our method with respect to model size using the Pythia model suite [Biderman et al., 2023], perform several ablation studies for individual components of our IDK objective, and verify that the general language modeling ability of IDK-tuned models does not degrade.

Our work can be extended in several ways. For example, since we do not rely on any labels of our training data used for IDK-tuning, we potentially apply our objective for next-token predictions where it might be ill-posed. Instead, we can perform lightweight filtering of relevant next-token predictions, such as named entities, focusing our objective more on factual next-token predictions. Also, IDK-tuning can be applied during pretraining from scratch, where our IDK objective could have interesting interactions with the acquisition of new knowledge during this stage.

# 7 Limitations

We note a few limitations of our proposed method. First, it requires a full pretraining of LMs on relatively large corpus. This of course is both highly computationally expensive and time-consuming. It is likely often the case that this kind of training cannot be conducted on typical academic lab resources, on a large enough model, in a reasonable amount of time.

Second, as discussed in Section 4.4, our method may slightly harm certain language skills, such as long text generation. Other downstream skills may be affected more significantly. We further discuss potential risk and biases in Appendix A.

# Acknowledgements

Roi Cohen and Gerard de Melo received funding from The Goldman Sachs Group, Inc., New York, NY, USA. Konstantin Dobler thanks the German Federal Ministry for Education and Research

(BMBF) through the project «KI-Servicezentrum Berlin Brandenburg» (01IS22092) and the European Laboratory for Learning and Intelligent Systems (ELLIS) PhD program for support. We further express our gratitude to the NeurIPS 2024 reviewers for their helpful comments.

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

## A  Impact

As discussed in Section 1, one of the main disadvantages of current LMs is their tendency to factually mislead the user by generating factual incorrect statements. Hence, the main impact of our work is to reduce such factual mistakes via our proposed method. Still, it is evident that this sort of approach can by no means completely eliminate hallucinations. It is important to stress that we propose a single method, not a system design for safe deployment of LLMs. In practice, we anticipate our method to be coupled with other checks and balances, forming a safe system.

Additionally, in this work, we use The Pile as a dataset to train models. The Pile is a web-crawled corpus, which likely harbors text reflecting various forms of biases. One impact of applying IDK-tuning is that the model may learn to answer in a biased way if this bias appears in its training data, while avoiding answers that rarely appear in its training data. This shows the need for more research on compiling high-quality training corpora.

## B   Computational Resources

For IDK-tuning of `Mistral-7B-v0.1`, we use Nvidia H100 or A100 GPUs depending on availability. For IDK-tuning `pythia-70m − 2.8B`, we use 1-4 Nvidia A6000 GPUs. For IDK-tuning of `bert-base-cased`, we use a single Nvidia A100 GPU.

## C   Questions Rephrasing

As mentioned in Section 3.2, for TriviaQA and PopQA, where the input is formed as a question, we reduce each of these input examples into a sentence completion task input, using GPT4. If we denote a random input question from one of these datasets by $x$, then our prompt to GPT4 is the following:

```
Please rephrase the following question as an input for a sentence
completion task.  For example:
```

```
For the question:  "Where was Michael Jackson born?", the sentence
should be:  "Michael Jackson was born in".
```

```
For the question:  "Who is Barack Obama's wife", the sentence
should be:  "The wife of Barack Obama is".
```

```
For the question:  "Where in England was Dame Judi Dench born?",
the sentence should be:
```

We found this prompt to be effective enough after manually testing it on a development set of a 45 examples.

