# OpenReview forum: "I Don't Know: Explicit Modeling of Uncertainty with an [IDK] Token"
_NeurIPS.cc/2024/Conference — NeurIPS 2024 poster_

### Official Review · Reviewer_Ny7r · 2024-07-12

**Soundness:** 3
**Presentation:** 4
**Contribution:** 3
**Rating:** 7
**Confidence:** 4

**Summary:**

This paper proposes a training-based confidence calibration method, named IDK-tuning, for improving LLM factuality. Specifically, a special `[IDK]` ("I Don't Know") token is added to the model's vocabulary and an objective is introduced which shifts some probability mass of wrong predictions to the `[IDK]` token during continued pretraining, thereby encouraging the model to express uncertainty explicitly. Results of the IDK-tuned models are reported on commonly used factual benchmarks, showing the potential of this method for reducing hallucination while only suffering a slight sacrifice in terms of knowledge recall.

**Strengths:**

* Confidence calibration and LM hallucination are both important topics and this paper connects them in an interesting way.
* The proposed IDK-tuning method is intuitive and well-motivated. While there have been papers on supervised finetuning for calibration, most of them focus on aligning the model with human demonstration (which requires annotation) or synthetic data (which incur extra cost). The direct adaptation of the training objective that incorporates uncertainty seems quite novel to my best knowledge, and results seem promising with much higher precision and only slightly lower recall, which is suited for the current generally over-confident LLMs.
* The authors perform extensive experiments on the scaling behavior and ablation for different components of the objective function.
* The authors do not overclaim their contribution. Singularities of the experiment results (e.g. `NaN`s in loss and collapsed recall from the `pythia` models) are mentioned and analyzed, and interesting insights are drawn from them.
* The writing is clear and the paper is easy to follow.

**Weaknesses:**

* For multiple-choice QA tasks, the precision improvement as well as the absolute values appear lower than those for the factual sentence completion tasks (comparing Table 1 and 2).  Given this, and that the multiple-choice QA tasks might be more applicable, it would be helpful if the authors could report more detailed results on each of the `lm-eval-harness` tasks, and investigate QA or other downstream tasks more carefully.
* Code or tuned model checkpoints are not provided, although some details about the settings and resources are mentioned in the paper.

**Questions:**

* Is it possible or beneficial to add multiple special tokens to express the different levels of uncertainty that are finer-grained, instead of only a single `[IDK]` token?
* For the evaluation on `TriviaQA` and `PopQA`, why do you reformat them into sentence completion tasks? Is it because in this way the next token(s) can be directly modeled without being extracted from a complete answer?

**Limitations:**

The authors highlight several limitations, including the need for full pretraining on large corpus, which is costly, and that the method can slightly sacrifice recall or the overall performance on some downstream tasks such as text generation.

---

> ### Author Rebuttal · Authors · 2024-08-07
>
> We sincerely thank the reviewer for their time reading our work, writing a thorough review and bringing up thoughtful comments.
>
> We are encouraged that the reviewer finds our method intuitive, novel and well-motivated, that our experiments are deemed extensive, and that the paper is considered well-written and as not overclaiming the results.
>
> Here are our responses to the weaknesses raised by the reviewer:
> - We agree with the reviewer that it would be beneficial to more carefully analyze and represent the lm-eval-harness results. In the camera-ready version, we will add the results of each of the datasets separately and discuss the difference between them to those of the other datasets. Importantly, we note that according to a comment by a different reviewer, we have conducted an in-depth analysis of our model’s mistakes. The setup of the analysis is the following:
> We randomly sample 200 examples (out of all the datasets) on which the IDK-tuned model generates a wrong answer (without predicting the [IDK] token). We then categorizes it to one of these four categories: No Effect (both the original model and our model generate the same answer), Noise (the original model knows it, while after our training it doesn’t), White Noise (both the original model and ours don’t know it, though they generate different answers), and Abstain (when our model abstains from answering while generating text like “unknown” or “a mystery”). For this analysis we take three different models: Mistral-7B, Pythia-2.8B and Pythia-70M. The results are the following:
>
> | model       | No Effect | Noise | White Noise | Abstaining |
> |-------------|-----------|-------|-------------|------------|
> | Mistral-7B  | 68.5      | 9     | 6.5         | 16         |
> | Pythia-2.8B | 59.5      | 13.5  | 12.5        | 14.5       |
> | Pythia-70M  | 52        | 18.5  | 22          | 7.5        |
>
> These results suggest that first, the bigger the model, the fewer changes our training approach causes in the model’s generations, and second, the bigger the model, the greater its ability to abstain from answering via words (which is generally equal to generating the new [IDK] token, though harder to evaluate automatically).
> - We will definitely provide the code and checkpoints with the camera-ready version. Additionally, all the information needed to reproduce our model is demonstrated in the paper.
>
>
> Here are our responses to the questions raised by the reviewer:
> - *“Multiple IDK tokens”*: In principle, the single [IDK] token does cover all levels of uncertainty, as we measure a continuous amount of probability mass put on the token. However, we think multiple different [IDK] tokens are interesting, e.g., discriminating between different “categories” of uncertainty such as lack of knowledge, lack of context etc.
> - *“TriviaQA / PopQA reformatting”*: Yes, we perform the reformatting to enable a straightforward evaluation.

---

> > ### Comment · Reviewer_Ny7r · 2024-08-10
> > **Official Comment by Reviewer Ny7r**
> >
> > Thanks for your response and additional results. I decide to keep my score unchanged.

---

### Official Review · Reviewer_4r2S · 2024-07-12

**Soundness:** 2
**Presentation:** 3
**Contribution:** 2
**Rating:** 3
**Confidence:** 4

**Summary:**

The paper proposes calibrating LLMs during a continued pertaining phase via an added [IDK] token to model uncertainty.

**Strengths:**

- The paper is structured well and written coherently.
- The introduction of the $\texttt{[IDK]}$ token to explicitly model uncertainty in LLMs is a novel approach
- Through ablation studies, the behavior of the proposed method is investigated extensively.

**Weaknesses:**

- **Theory**:
  - The approach is mathematically not very well grounded. Also, mathematical expressions such as $prob(\texttt{[tok]}, \hat{y})$ and $prob(argmax(\hat{y}), \hat{y})$ do not cohere with common practices [1,2,3,4]. The authors could consider something like $p(y_{t}=\texttt{[tok]} | y_{<t}, x)$ and $max_{i} \ p(y_{t}=i | y_{<t}, x)$.
  - The uncertainty factor is only bigger than $0$ if any other token gets assigned a higher probability than the $\texttt{[gold]}$ token. It does not account for the case where a model is uncertain about *any* token and thus predicts a (low) probability for all tokens. For instance, consider the tokens relating to ($\text{"Paris"}$, $\text{"Berlin"}$, $\text{"London"}$, $\text{"Rome"}$, $\text{"Vienna"}$). If the model predicts any of $p(y_{t} | \text{"The capital of France is"}) \in [(0.2, 0.2, 0.2, 0.2, 0.2), (0.3, 0.1, 0.2, 0.2, 0.2), ...]$, the uncertainty factor is $0$ no matter the hyperparameter $\Pi$, while it is clear that in all those cases the model is uncertain about the correct next token. The probability of the $\texttt{[IDK]}$ gets even decreased via the uncertainty regularization.

- **Evaluation**: The authors do not compare against other uncertainty quantification methods, such as (length-normalized) predictive entropy [1], p(true) [2], or semantic entropy [3,4]. These methods do not require additional pertaining, and thus do not suffer from training instabilities, mode collapse, or high computational costs, but can directly be applied to "off-the-shelf" models. Additionally, there exist methods that consider fine-tuning models to express their lack of knowledge that have not been considered.


---
[1] A. Malinin and M. Gales. Uncertainty estimation in autoregressive structured prediction.

[2] S. Kadavath, T. Conerly, A. Askell, T. Henighan, D. Drain, E. Perez, N. Schiefer, Z. Hatfield-Dodds, N. DasSarma, E. Tran-Johnson, S. Johnston, S. El-Showk, A. Jones, N. Elhage, T. Hume, A. Chen, Y. Bai, S. Bowman, S. Fort, D. Ganguli, D. Hernandez, J. Jacobson, J. Kernion, S. Kravec, L. Lovitt, K. Ndousse, C. Olsson, S. Ringer, D. Amodei, T. Brown, J. Clark, N. Joseph, B. Mann, S. McCandlish, C. Olah, J. Kaplan. Language Models (Mostly) Know What They Know.

[3] L. Kuhn, Y. Gal, and S. Farquhar. Semantic uncertainty: Linguistic invariances for uncertainty estimation in natural language generation.

[4] L. Aichberger, K. Schweighofer, M. Ielanskyi, and S. Hochreiter. Semantically Diverse Language Generation for Uncertainty Estimation in Language Models.

**Questions:**

- If an IDK-tuned model is to be aligned, how does the alignment hurt IDK-tuning? Also, if a model is IDK-tuned after alignment, how does the pertaining hurt alignment?
- The method is evaluated on a single model. How do you guarantee that the results generalize to bigger models?

**Limitations:**

The authors adequately addressed the limitations.

---

> ### Author Rebuttal · Authors · 2024-08-07
>
> We are very thankful to the reviewer for their time reading our work and writing a thorough review.
>
> We are encouraged that the reviewer finds our paper to be well written, our approach to be novel, and our ablation experiments to be extensive and to properly demonstrate the behavior of our approach.
>
> We will now address the reviewer's concerns:
>
> ## Theory
> - For the camera-ready version, we will modify all of the mathematical expressions in the paper for better coherence with previous common practices. Thank you for the suggestions!
> - *“The uncertainty factor is only bigger than $0$ if any other token gets assigned a higher probability than the $\texttt{[gold]}$ token”*: We first thank the reviewer for this comment as well, which we strongly agree with. As each of the training runs is very resource-intensive, this idea has not been evaluated in a proper experimental setup. We plan to test a model that has been trained with a slight modification to our proposed training objective, reconsidering the true certainty it has on the gold token, even if this is the maximal probability token among the others. For example, we could take “1 - P(gold token)” as an additional term to be combined in the “Uncertainty Factor” definition, as well as to remove the separation we apply between cases where this factor is zero and the others, in terms of our objective. We will add some initial results of this alternative objective in the camera-ready version. We additionally think this is definitely an exciting follow-up to our work.
>
> ## Evaluation
> Thank you for this comment. We also believe that evaluating other uncertainty quantification methods might strengthen our work. We thus decided to add the results of the methods suggested by the reviewer to our paper. We will now provide the results of the “P(true)” and the “semantic entropy” methods using the Mistral-7B model, and will add these together with the results of the other models to our camera-ready version of the paper. For the “semantic entropy” method we use SOTA named entity recognition in order to extract only the “answer” itself from the model’s generation. It is important to note though that our method could potentially be applied directly during the normal pretraining of the model, by only modifying the pretraining objective to be ours, and thus the claim that our method requires additional training would not apply. Moreover, the “Semantic Entropy” method indeed does not require any additional training, though it requires significantly more inference calls and some post-processing of the generations (clustering etc.).
> The results are the following:
> - **LAMA Google-RE**:
>
> | model                               | Precision | Recall | F1   |
> |-------------------------------------|-----------|--------|------|
> | Mistral-7B                          | 48.1      | 48.1   | 48.1 |
> | Mistral-7B + The Pile               | 48.8      | 48.8   | 48.8 |
> | Mistral-7B + Confidence Threshold   | 60.0      | 40.0   | 48.0 |
> | Mistral-7B + P(true)                | 54.4      | 44.5   | 48.9 |
> | Mistral-7B + Semantic Entropy       | 70.1      | 38.9   | 50.0 |
> | Mistral-7B + IDK Tuning on The Pile | 71.1      | 40.6   | 51.7 |
>
> - **LAMA T-Rex**:
>
> | model                               | Precision | Recall | F1   |
> |-------------------------------------|-----------|--------|------|
> | Mistral-7B                          | 71.2      | 71.2   | 71.2 |
> | Mistral-7B + The Pile               | 69.9      | 69.9   | 69.9 |
> | Mistral-7B + Confidence Threshold   | 80.4      | 63.5   | 71.0 |
> | Mistral-7B + P(true)                | 73.8      | 65.1   | 69.2 |
> | Mistral-7B + Semantic Entropy       | 88.0      | 65.4   | 75.0 |
> | Mistral-7B + IDK Tuning on The Pile | 88.5      | 65.5   | 75.3 |
>
> - **LAMA SQuAD**:
>
> | model                               | Precision | Recall | F1   |
> |-------------------------------------|-----------|--------|------|
> | Mistral-7B + P(true)                | 54.9      | 41.0   | 46.9 |
> | Mistral-7B + Semantic Entropy       | 70.2      | 44.5   | 54.4 |
> | Mistral-7B + IDK Tuning on The Pile | 72.0      | 44.3   | 54.9 |
>
> - **TriviaQA**:
>
> | model                               | Precision | Recall | F1   |
> |-------------------------------------|-----------|--------|------|
> | Mistral-7B + P(true)                | 58.8      | 47.5   | 52.5 |
> | Mistral-7B + Semantic Entropy       | 68.5      | 52.5   | 59.4 |
> | Mistral-7B + IDK Tuning on The Pile | 72.5      | 52.0   | 60.6 |
>
> - **PopQA**:
>
> | model                               | Precision | Recall | F1   |
> |-------------------------------------|-----------|--------|------|
> | Mistral-7B + P(true)                | 40.3      | 29.0   | 33.7 |
> | Mistral-7B + Semantic Entropy       | 68.7      | 20.4   | 31.5 |
> | Mistral-7B + IDK Tuning on The Pile | 78.1      | 20.5   | 32.5 |
>
> These results suggest that our approach still leads to the best precision and f1 scores compared to the new baselines too, though the gaps are smaller compared to the ones against the previous baselines.
>
> ## Questions
> - *"Combining IDK-tuning with alignment”*: While we do not explicitly study this in our work, we do believe it is an interesting direction for future work. Specifically, the line of work on Task Arithmetic [1] will be interesting: Can we extract and combine the “IDK” weight vector with an instruction-tuned model?
> - *"“The method is evaluated on a single model” / “generalization to bigger models”*: We believe this question must be a misunderstanding, as we conduct very extensive experiments with eight different models, spanning encoder-only (BERT) and decoder-only architectures, as well as an explicit study of scaling behavior using the Pythia model suite and Mistral-7B.
>
> In light of this, we kindly request you to reconsider and appropriately raise the score of your review if we have sufficiently addressed some or all of your concerns.

---

> > ### Comment · Reviewer_4r2S · 2024-08-08
> > **Theoretical and Performance Concerns**
> >
> > Thank you for the rebuttal.
> >
> >
> > > *Review*: **The uncertainty factor is only bigger than 0 if any other token gets assigned a higher probability than the **[gold]** token. It does not account for the case where a model is uncertain about any token and thus predicts a (low) probability for all tokens.**
> >
> > > *Rebuttal*: We first thank the reviewer for this comment as well, which we strongly agree with. As each of the training runs is very resource-intensive, this idea has not been evaluated in a proper experimental setup.
> >
> > The main contribution of your work is the introduction of an objective function that shifts the probability mass to the **[IDK]** token for incorrect predictions. This is not just an "idea" from my perspective; it reveals a significant theoretical flaw in your work.
> >
> > Given that your method lacks a solid theoretical foundation and the empirical performance only shows marginal improvements over the current state-of-the-art uncertainty quantification methods, coupled with major drawbacks such as unknown behavior when combined with alignment, I must reject the paper at this point.
> >
> > Addressing the theoretical issues will presumably enhance the performance of your method. Also, insights into how your method can be effectively applied to instruction-tuned models will improve the work.

---

> > > ### Author Response · Authors · 2024-08-10
> > >
> > > Thank you for your quick response!
> > >
> > > *Regarding the theoretical issue that has been mentioned*:
> > >
> > > The behavior the reviewer describes as a flaw is actually what we aim for - as long as the model "knows" the answer, namely its maximal token predication is correct, we want to encourage it and raise even more its confidence on it, while also decreasing its "uncertainty" (the probability it puts on the [IDK] token). Thus, we claim that the fact that the uncertainty factor is bigger than 0 only if any other token gets assigned a higher than the gold token is not actually a theoretical flaw – enabling this would provide a signal to predict [IDK] when the model does in fact know the answer. Therefore, implementing this in the way the reviewer suggested is also valid but would create a subtly different objective which stands for a subtly different goal.
> > > Additionally and very importantly – the semantic entropy method for calibration requires way more inference calls (might be more than 10 times more), and an external additional clustering method run. We think this is an important point to consider while looking at the results – even though our method is only "marginally" better, it is applied once during pretraining and doesn't add any more computation effort during inference time at all.
> > > To sum up, we will extend the introduction to discuss these points. We do believe that in practice our technique proves useful to reduce hallucination, which is a very important societal challenge in dire need of further advances.
> > >
> > > *Regarding the alignment point mentioned by the reviewer*:
> > >
> > > It is well-known that alignment techniques disrupt the probability estimates delivered by large language models, so this is not an issue unique to our specific paper. Our paper suggests a language modeling objective that encourages uncertainty expression via a new [IDK] token, and thus we believe it should be applied during or immediately after the initial pretraining. However, we do believe that our method could be complemented with further techniques for better post-alignment uncertainty modeling. We believe this is an exciting line of future work, and we will add this point to our discussion section.

---

> > > > ### Comment · Reviewer_4r2S · 2024-08-14
> > > >
> > > > Thank you for the comment.
> > > >
> > > > > The behavior the reviewer describes as a flaw is actually what we aim for -  as long as the model "knows" the answer, namely its maximal token predication is correct, we want to encourage it and raise even more its confidence on it, while also decreasing its "uncertainty" (the probability it puts on the [IDK] token). Thus, we claim that the fact that the uncertainty factor is bigger than 0 only if any other token gets assigned a higher than the gold token is not actually a theoretical flaw – enabling this would provide a signal to predict [IDK] when the model does in fact know the answer. Therefore, implementing this in the way the reviewer suggested is also valid but would create a subtly different objective which stands for a subtly different goal.
> > > >
> > > > I understand your objective of reinforcing the model’s confidence when it "knows" the answer. However, in scenarios where the entropy of the predictive distribution is high (as in the example I provided above), the model does **not** "know" of the answer. Yet, the uncertainty factor remains 0, leading to a reduction in the probability assigned to the *[IDK]* token. My intention is not to propose an alternative solution, but to highlight an unintended behavior that does not align with your intended goal.
> > > >
> > > > > It is well-known that alignment techniques disrupt the probability estimates delivered by large language models, so this is not an issue unique to our specific paper.
> > > >
> > > > You are correct that alignment techniques generally disrupt probability estimates in large language models, making this a broader issue. However, unlike other uncertainty measures, your method relies on this behavior to estimate the uncertainty of aligned models. Given the importance of this use case, I believe it is essential to address this concern more thoroughly in your work.
> > > >
> > > > I maintain that addressing these two points is crucial.

---

### Official Review · Reviewer_i3GV · 2024-07-12

**Soundness:** 3
**Presentation:** 3
**Contribution:** 3
**Rating:** 6
**Confidence:** 4

**Summary:**

To allow LMs to express their uncertainty for generative tasks, the authors introduce a new special IDK token. The authors modify the cross-entropy training objective to assign part of the probability mass to the IDK token in cases where the model gets the prediction wrong. The token embedding is randomly initialized and then refined through additional fine-tuning of pretrained LLMs of various sizes and types (Pythia, Mistralv1, BERT).
Through experiments on a range of completion, QA & MCQA datasets the authors show that IDK tuning positively affects precision at a slight cost of recall. The authors perform further ablation experiments solidifying their choices for the loss weight hyperparameter as well as the regularization term. The authors further show that IDK tuning does not significantly adversely affect other capabilities of the underlying LMs.

**Strengths:**

- Well written and easy to follow
- To the best of my knowledge, presents a novel approach to quantifying uncertainty in LMs
- Exhaustive experimental evaluation & ablation over different hyperparameter choices demonstrating robustness of the proposed approach

**Weaknesses:**

- The IDK-tuning setup requires sometimes prohibitive additonal fine-tuning of the base LM
- It is unclear how the method would be applied when a model would be pretrained from scratch with the IDK token included - it is natural that LMs will be worse in predicting tokens as initial stages of training, so the pretrain - add IDK - tune paradigm seems as the only current option. The two previous points slightly limit the applicability of the metod.
- While the reported F scores are generally higher compared to baselines and alternatives, the IDK-tuned models still suffer from tangibly lower recall.

**Questions:**

None

**Limitations:**

Yes

---

> ### Author Rebuttal · Authors · 2024-08-07
>
> We first express our sincere gratitude to the reviewer for their time reading our work, writing a thorough review, and bringing up thoughtful comments.
>
> We are encouraged that the reviewer finds our paper to be well written, our approach to be novel, and our experiments to be extensive and to properly demonstrate the robustness of our approach.
>
> Here are our responses to the concerns raised by the reviewer:
> - While our method does require finetuning, we also benefit from the vast amount of current work in the optimization and efficiency of LLM training. In any case, we argue that (moderate) finetuning does not constitute a technical weakness of our method.
> - While we experiment with IDK-tuning for the reasons you mention, we believe integrating the [IDK]-flavor of uncertainty-aware training into pretraining from scratch presents an exciting direction for future work (e.g., via loss weight schedules). We do believe that with small adaptations of our loss function, our training method could be applied for pretraining from scratch. One of the main reasons we couldn’t check this effectively is the very extensive resources (time, compute and memory) these controlled experiments in this setting would have consumed (which would be impractical or impossible on typical academic budgets in a reasonable amount of time). Also, we believe this direction of work warrants a detailed study that will fill another whole paper.
> - We agree that our method is not “free” (as it leads to somewhat lower recall), but argue that it presents a valuable tradeoff, given the importance of reducing hallucinations.

---

> > ### Comment · Reviewer_i3GV · 2024-08-12
> >
> > Thank you for the response. I will keep my score unchanged as it still accurately reflects my feelings on the paper.

---

### Official Review · Reviewer_aDMj · 2024-07-15

**Soundness:** 2
**Presentation:** 3
**Contribution:** 2
**Rating:** 4
**Confidence:** 4

**Summary:**

It introduces a novel method to address the issue of hallucinations in Large Language Models (LLMs). These models, despite their proficiency in capturing and generating human knowledge, can sometimes produce factually incorrect text. To combat this, the authors propose a calibration method that incorporates an [IDK] ("I don’t know") token into the model's vocabulary.

**Strengths:**

The introduction of the [IDK] token is a creative solution to an existing problem in LLMs. It represents a novel way to handle uncertainty, which is not just a new definition or problem formulation but also a practical application within language models.

The paper proposes a new pretraining objective that modifies the conventional cross-entropy loss to incorporate the [IDK] token. This is an original contribution to the field of natural language processing.

**Weaknesses:**

The paper primarily uses The Pile for training, which may not be representative of all possible language use cases or domains.

While the paper provides a good overview of the performance metrics, an in-depth error analysis could offer more insights into the types of errors the models are making and how the [IDK] token impacts these.

As the model is trained on web-crawled data, there is a risk of learning and perpetuating societal biases present in that data.

**Questions:**

See Weaknesses

**Limitations:**

While the paper notes the potential for bias in the training data, it could provide more details on how this might affect the model's predictions and decision-making.

The paper could more explicitly discuss the potential for the model to contribute to the spread of misinformation, especially if it fails to correctly identify uncertain or incorrect information.

---

> ### Author Rebuttal · Authors · 2024-08-07
>
> We first highly thank the reviewer for their time reading our work, writing a thorough review and bringing up thoughtful comments.
>
> We are encouraged that the reviewer finds our method a creative and novel method to handle an existing and important problem current LLMs have - which is generating misinformation and hallucinations.
>
> We would like to address the following weaknesses / limitations of our work raised in the review:
> - **“Using primarily The Pile”**: We argue that using a general, English-centric dataset is well suited to demonstrate the efficacy of our method and not an inherent weakness of our method. Additionally, as you may assume, each of these training sessions is extremely resource-intensive. Exploring other languages and domains is a good direction for future work.
> - **“In-depth error analysis missing”**: We do provide a very detailed analysis and ablation of precision-recall tradeoffs in Section 4.2. Based on the reviewer’s suggestion, we decided to conduct an in-depth error analysis of our model’s mistakes. We will provide the results here and will add them to the camera-ready version of the paper too. The setup of the analysis is the following:
> We randomly sample 200 examples (out of all the datasets) on which the IDK-tuned model generates a wrong answer (without predicting the [IDK] token). We then categorize these to one of four categories: No Effect (both the original model and our model generate the same answer), Noise (the original model knows it, while after our training it doesn’t), White Noise (both the original model and ours don’t know it, though they generate different answers), and Abstain (when our model abstains from answering while generating text like “unknown” or “a mystery”). For this analysis we take three different models: Mistral-7B, Pythia-2.8B and Pythia-70M. The results are the following:
> | model       | No Effect | Noise | White Noise | Abstaining |
> |-------------|-----------|-------|-------------|------------|
> | Mistral-7B  | 68.5      | 9     | 6.5         | 16         |
> | Pythia-2.8B | 59.5      | 13.5  | 12.5        | 14.5       |
> | Pythia-70M  | 52        | 18.5  | 22          | 7.5        |
>
> These results suggest that first, the bigger the model, the fewer changes our training approach causes in the model’s generations, and second, the bigger the model, the greater its ability to abstain from answering via words (which is generally equal to generating the new [IDK] token, though harder to evaluate automatically).
> - **“Bias from web-crawled pretraining data”**: We agree and will use the extra page to discuss this in an extended Limitations section. However, while important to consider, we argue that pretraining data bias is (1) mostly inherent to the way LLMs are trained nowadays and (2) not a particular weakness of *our method* but an exciting direction for follow-up work (e.g., using synthetic data).
> - **"Potential for the model to contribute to the spread of misinformation"**: Indeed our extended Limitations section will also discuss the risk of misinformation. It is important to stress that we propose a single *method*, not a *system design* for safe deployment of LLMs. In practice, we anticipate our method to be coupled with other checks and balances, forming a *safe system*.
>
> In light of this, we kindly request you to reconsider and appropriately raise the score of your review if we have sufficiently addressed some or all of your concerns.

---

### Author Rebuttal · Authors · 2024-08-07

We thank all reviewers for their time and effort put into providing a thorough review of our work. We briefly highlight the strengths of our work as identified by the reviewers:
- Our [IDK] token approach is a “novel approach” deemed “an original contribution to the field of natural language processing”, “well-motivated”, and “creative”.
- The reviewers have praised our extensive experiments and ablation studies, while not overclaiming our results.
- Our paper is well-written and easy to follow.

In the individual responses, we address all points and questions raised by the reviewers in detail.

---

### Decision · Program_Chairs · 2024-09-25

**Decision:**

Accept (poster)

**Comment:**

The paper proposes a continued pretraining strategy, assuming a model has been initialized, does another round of pretraining where for incorrect (next token) predictions, the loss function remaps some of the probability mass to an [IDK] token (instead of trying to push the model to be correct).

The empirical results are positive and the authors have done more baseline comparisons (see rebuttal to Reviewer 4r2S)

Reviewers i3GV and Ny7r are positive about the paper. The main concern about the paper is regarding the mathematical definition of the loss function (Reviewer 4r2S). While I agree that the reviewer's formulation may be more elegant I don't think the authors formulation is wrong i.e. the question is whether you only allocate probability mass to the [IDK] token when the model is incorrect (what the authors did) or if the overall uncertainty is high regardless of whether it is correct (what the reviewer is advocating for)

Therefore I favor accepting the paper.